# Spatiotemporal patterning of photoresponsive DNA-based hydrogels to tune local cell responses

Fujian Huang [1✉], Mengxi Chen[1], Zhixin Zhou[2], Ruilin Duan[1], Fan Xia [1✉] & Itamar Willner [2✉]

Understanding the spatiotemporal effects of surface topographies and modulated stiffness and anisotropic stresses of hydrogels on cell growth remains a biophysical challenge. Here we introduce the photolithographic patterning or two-photon laser scanning confocal microscopy patterning of a series of o-nitrobenzylphosphate ester nucleic acid-based polyacrylamide hydrogel films generating periodically-spaced circular patterned domains surrounded by continuous hydrogel matrices. The patterning processes lead to guided modulated stiffness differences between the patterned domains and the surrounding hydrogel matrices, and to the selective functionalization of sub-regions of the films with nucleic acid anchoring tethers. HeLa cells are deposited on the circularly-shaped domains functionalized with the MUC-1 aptamers. Initiation of the hybridization chain reaction by nucleic acid tethers associated with the continuous hydrogel matrix results in stress-induced ordered orthogonal shape-changes on the patterned domains, leading to ordered shapes of cell aggregates bound to the patterns.

[1] State Key Laboratory of Biogeology and Environmental Geology, Engineering Research Center of Nano-Geomaterials of Ministry of Education, Faculty of Materials Science and Chemistry, China University of Geosciences, Wuhan, China. [2] Institute of Chemistry, Center for Nanoscience and Nanotechnology, The Hebrew University of Jerusalem, Jerusalem, Israel. ✉email: huangfj@cug.edu.cn; xiafan@hust.edu.cn; itamar.willner@mail.huji.ac.il

Substantial continuous research efforts are directed towards the development of soft polymer matrices or water swollen polymer networks (hydrogels mimicking the physiochemical properties of tissues[1,2]). Different biomedical applications of these soft materials were suggested, including stimuli-responsive matrices for controlled drug release[3–8], signal-triggered prosthetic elements[9–12], the fabrication of medical devices[13–15], and tissue engineering[16,17], where the search for extracellular matrix component (ECM) alternatives is a major effort. The ECM composite is a complex blend of proteins, glycoproteins, proteoglycans and more, and represents a non-homogeneous dynamic environment that controls cell adhesion, migration, and proliferation[18,19]. Mimicking such interactions between cells and synthetic polymers/hydrogels is difficult due to the homogeneous physiochemical properties of two-dimensional or three-dimensional polymer/hydrogel/matrices. The adhesion and locomotion of cells on hydrogels are controlled by the elasticity and stiffness of the hydrogels[20,21] and the shapes of the adsorbed cells are influenced by ligands associated with the hydrogels[22,23].

Also, the topography of the surfaces affects the alignment, migration, and differentiation of the cells due to the anisotropic stresses that presumably affect the cytoskeletal organization and cell morphology[24,25]. In addition, cell cultures are strongly affected by the three-dimensional environment of hydrogel matrices. Biochemical modulation and physical tunability of bulk hydrogel properties modulated in space and time (e.g., spatiotemporal), such as stiffness, water content, or molecule diffusivity are controlled by the crosslinking of the hydrogel network, and lead to spatiotemporal heterogeneity of the hydrogels, mimicking native ECM[26]. For example, exposure of hydrogels to light may introduce heterogeneity into hydrogel matrices by spatial, time-controlled (dose) irradiation of the material leading to crosslinking variations and to patterned surfaces that result in predesigned hydrogel topographies of dictated stiffness, thus providing anisotropic environments for cell adhesion, motility and dictated proliferation[27,28].

Indeed, different approaches were used to photochemically control the stiffness of soft polymers or hydrogel matrices. These include the light induced crosslinking of the polymer networks by dimerization[29], addition[30] or cycloaddition between functional groups associated with the networks[31–33]. For example, dimerization of acrylate groups, photoinduced Michael addition of thiolates to double bonds[34–36] or symmetry-allowed 2 × 2 cycloadditions were used to crosslink hydrogel matrices. Alternatively, the tethering of photoisomerizable units such as azobenzene[37–39] or nitrospiropyran units[40] to hydrogels were used to control the hydrophilicity, swelling degree and molecule diffusivity[41,42]. Indeed, different photoresponsive polymers and hydrogels[43–45] revealing switchable interactions with cells, patterning of biomolecules on polymer matrices and their controlled release[46,47] were demonstrated. A different approach to photocontrol the crosslinking of hydrogels involves the photocleavage of crosslinking bridging units. For example, the photocleavage of o-nitrobenzyl-functionalized bridging units provides a means to decrease the levels of crosslinking[48]. Despite the progress in advancing the photochemical control over the spatiotemporal properties of hydrogels, there is no versatile material to design and master the photochemical, on-command, programmable spatiotemporal heterogeneity of hydrogels.

The base sequence comprising nucleic acids provides substantial structural and functional information that can be dictated by auxiliary chemical or physical triggers (pH[49,50], ions[51,52], strand displacement[53,54], heat[55], light[56]) thereby controlling switchable material properties[57]. These versatile features of nucleic acids were used to modify synthetic polymers (polyacrylamide[58], carboxymethyl cellulose[59]) and to develop stimuli-responsive DNA-based hydrogels that revealed triggered switchable stiffness properties[60]. Different applications of these hydrogels were demonstrated, including programmable shape changes[61], shape memory[62], self-healing[63], switchable catalysis[64], and controlled drug release[65]. In addition, the patterning of stimuli-responsive DNA-based hydrogel films[66], and the fabrication of stimuli-responsive microstructures, e.g., microcapsules[67] were reported. Also, o-nitrobenzylphosphate ester photoprotected nucleic acids were applied for the photolithographic patterning of microparticles.[68] Different applications of photoresponsive o-nitrobenzylphosphate modified nucleic acid were demonstrated including the fabrication of photoresponsive microcapsules for controlled drug release[69], the patterning of DNA modified surfaces[70,71], pattern transformation in hydrogels[72], and photo-controlled hybridization in hydrogels[73].

In the present study we introduce a series of polyacrylamide hydrogels functionalized with different o-nitrobenzylphosphate ester-modified nucleic acid modules for the light induced patterning of hydrogel matrices that result in functional programmed spatially ordered patterned domains. The photopatterning is achieved by photolithography through masks or by two-photon laser scanning confocal microscope. Two-dimensional and three-dimensional patterning of the hydrogels are demonstrated. The patterned hydrogel films are spatially modified with nucleic acid tethers that are used for the ordered association of complementary nucleic acids or for the targeted deposition and growth of cells using aptamer ligand interactions[74]. In addition, the photopatterning of a hydrogel with a periodically spatially-separated circular hole containing array is presented. The tethers linked to the crosslinking units of hydrogel act as promoter strands to activate the hybridization chain reaction (HCR)[66,75] coating of the hydrogel, leading to its swelling and the formation of a mechanical stress within the hydrogel. A secondary periodical orthogonal stress-induced deformation of the circular hole array into ellipsoid microstructures is demonstrated. In addition, the spatial deposition of HeLa cells in the hole-containing patterned array is presented, and the proliferation of the cells in the confined array of circular and ellipsoidal domains is discussed.

While many studies addressed the photopatterning of hydrogels and the control over their stiffness[43–47], the present study introduces a unique common approach to pattern hydrogel matrices by versatile combinations of o-nitrobenzylphosphate ester photoprotective units. This leads to a universal set of patterned materials with guided functionalities and programmed domains of variable stiffness. These diverse features of the patterned structures will be emphasized for the different systems. The design of variable-stiffness patterned domains in a continuous hydrogel matrix, and the possibility to introduce specific cell-binding agents into the patterned domains provide versatile means to selectively associate cells to the domains and to examine neighboring matrix effects on the cell growth.

## Results and discussion

**Structures of the photoresponsive hydrogel films**. Figure 1 outlines the different structures of the photoresponsive (o-nitrobenzylphosphate ester) nucleic acid based polyacrylamide copolymer hydrogels used in the present study. The hydrogels consist of polymerized acrylamide and methylacrylamide derivatives selected from the library outlined in Fig. 1a composed of the self-complementary nucleic acid duplex (a)/(a) functionalized at its 5′-ends with acrydite monomer units, (1), the nucleic acid strand (b) modified at its 5′-end with the o-nitrobenzylphosphate ester photoresponsive group linked to the acrydite monomer, (2), the hairpin nucleic acid (c) containing in its loop region the

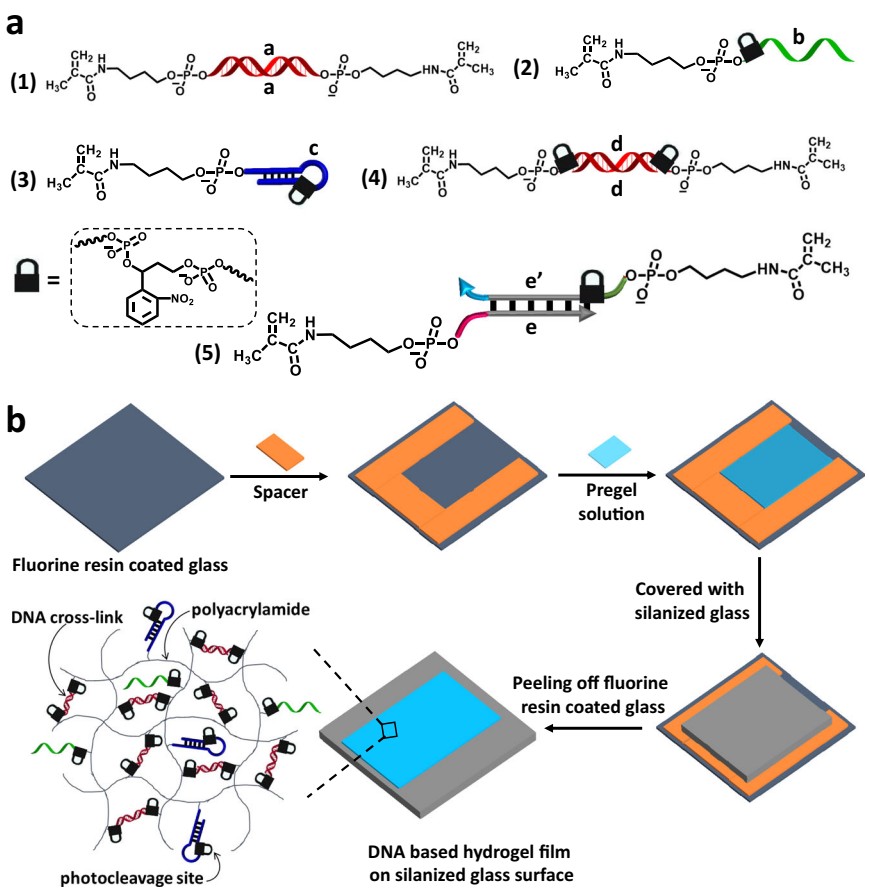

**Fig. 1 Structures of the acrydite modified DNA strands used to construct the hydrogels and sequential steps to prepare the hydrogels. a** Schematic structure composition comprising the photoresponsive o-nitrobenzylphosphate esters constituents that build the light-sensitive polyacrylamide hydrogels. Constituents to construct the diverse light-responsive hydrogels include: (1) Acrydite-modified self-complementary duplex (a)/(a). (2) Acrydite-modified ortho-nitrobenzylphosphate ester protected strand (b). (3) Acrydite-modified ortho-nitrobenzylphosphate ester protected hairpin (c). (4) Acrydite-functionalized ortho-nitrobenzylphosphate ester self-complementary duplex (d)/(d). (5) Acrydite-functionalized orthonitrobenzylphosphate ester modified complementary strands comprising the duplex (e)/(e'). **b** Stepwise synthesis of the hydrogel films that include the possible and variable acrydite-modified units (1)-(5) as possible components of the hydrogel.

o-nitrobenzylphosphate ester photoresponsive unit and tethered through its 5′-end to the acrydite monomer, (3), the self-complementary nucleic acid duplex (d)/(d) modified at their 5′-ends with the photoresponsive o-nitrobenzylphosphate ester groups and tethered to the acrydite monomer, (4), and the duplex nucleic acids (e)/(e') that includes at the 5′-end termini of (e') the o-nitrobenzylphosphate ester group and at phosphorylated 5′-ends of (e) and (e') a covalently-tethered to acrydite unit, (5). The hydrogels consist always of the acrylamide monomer and of two or more of the acrydite derivatives comprising the library. Note that the bis-acrydite derivatives (1), (4) and (5) act as crosslinkers of the resulting hydrogel matrices, the monomers (2) and (3) are introduced into the polyacrylamide chains without crosslinking capacities.

The formation of the hydrogel films is schematically outlined in Fig. 1b. A spacer was mounted on a fluorine resin coated glass slide and the respective pre-gel mixture was placed within the spacer domain and covered with a counter silanized glass slide. After polymerization, the polymerized film was transferred to the silanized glass slide and subjected to photolithographic patterning through a "hole" containing mask ($\lambda = 365$ nm) or by a light-induced patterning process applying a two-photon laser scanning confocal microscope, $\lambda = 740$ nm. It should be noted that the patterning of o-nitrobenzylphosphate photoprotected hydrogel film can be affected by the diffusibility of the degraded

photoproducts. This issue was previously addressed and characterized by the attenuation of the light source[76,77]. Nonetheless, the very thin thickness of the film along the z-dimension suggest that such perturbations of the patterning process should be negligible. In the next sections we will apply the versatile photopatterning procedure to design methods to control the stiffness of the irradiated/non-irradiated domains of the hydrogel films and to provide means to introduce by the photolithographic method specific functional ligands into the irradiated/non-irradiated domains.

**Photopatterned tethering of functional DNA strands within hydrogel film**. The first system that uses the photocleavable nucleic acid modules to generate the patterned hydrogel film is depicted in Fig. 2. The hydrogel is composed of the polymerized modules (1) and (2), where (1) provides the crosslinking elements and (2) yields the monomer comprising the nucleic acid tether chains. Irradiation of the hydrogel film through a "hole" containing mask leads to the photocleavage of the photoprotective groups linking the nucleic acid tether of (2) to yield the (1)-crosslinked hydrogel backbone in the irradiated areas, while the hydrogel domains protected from the irradiation remain intact (crosslinked by (1) and functionalized with (2)). Removal of the mask and rinsing of the hydrogel film, followed by the

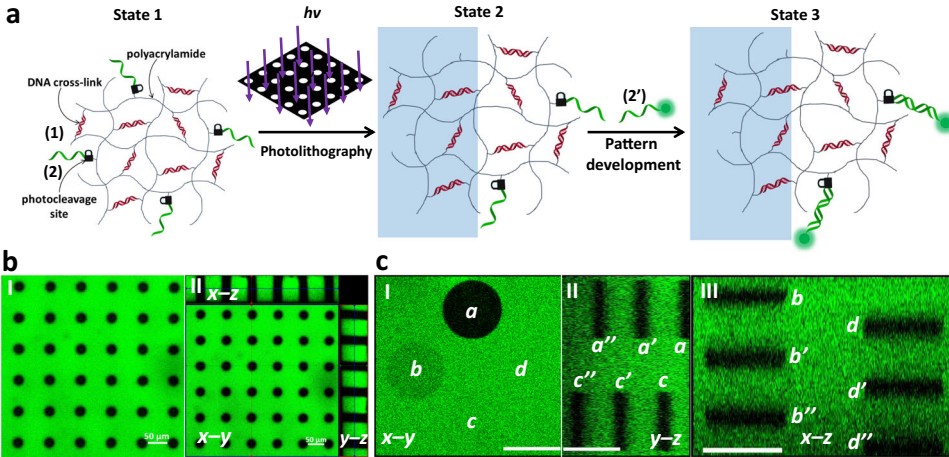

**Fig. 2 Light-induced patterning and imaging of a photoresponsive polyacrylamide hydrogel film functionalized with (1) and the o-nitrobenzylphosphate ester nucleic acid (2). a** Photolithographic patterning of the hydrogel through a circular-hole containing mask ($\lambda = 365$ nm), and the fluorescence imaging of the resulting pattern by the hybridization of fluorescein labeled (2') with the tethers (2) associated with the non-irradiated domains. **b** Panel I—Fluorescence confocal microscope imaging of the patterned hydrogel. Panel II—Three-dimensional confocal microscope imaging of the hydrogel film across the x-y, x-z, and y-z planes. Scale bars are 50 μm. **c** Fluorescence confocal microscope imaging of twelve layered and positioned circular domains generated by two-photon laser scanning confocal microscope lithographic patterning process ($\lambda = 740$ nm) using different spatial and positional focusing depths. Panel I—Confocal microscope imaging of the x-y plane consisting of four rectangular circular patterns generated at different focal depths. Panel II—Confocal microscope imaging of the two-photon laser patterned hydrogel film across the y-z plane where the imaging microscope is focused at patterned columns **a** and **c**. Panel III—Confocal microscope imaging of the two-photon laser patterned hydrogel film across the x-z plane where the imaging microscope is focused at patterned columns **b** and **d**. Scale bars correspond to 20 μm. Realizing that the patterned films reveal a thickness of ca. 80 μm, the UV light penetrates across the entire z dimension of the thin film layer, and thus the depth profile of the patterns correspond to the film thickness. This is supported by the depth of the optical images of the patterns that coincide with the macroscopic thickness of the film. $N = 4$ independent experiments were performed.

hybridization of the fluorescein-modified nucleic acid (2') with the free tethers of (2) associated with the non-irradiation domains of the hydrogel, yields the patterned hydrogel film shown in Fig. 2a composed of a continuous green fluorescent area of the fluorescein labels and "dark" non-fluorescent circles of the irradiated pattern as shown in the fluorescence confocal microscope image (Fig. 2b, panel I). The orthogonal confocal microscope imaging of the patterned hydrogel, Fig. 2b panel II, demonstrates the three-dimensional formation of patterned round cylinders across the hydrogel film.

Furthermore, the maskless photopatterning of the hydrogel film was accomplished by two-photon laser scanning confocal microscope lithographic patterning of the film ($\lambda = 740$ nm), Fig. 2c. In this experiment, the focal depth and spatial position of the laser are positionally fixed, and thus, the patterned domains across the x-y-z axes can be controlled. The three-dimensional hydrogel film is patterned by twelve circular domains along the x-y-z axes of the hydrogel film at twelve different focal positions of the x-y-z axes. In the first focal x-y plane, one circle is patterned. Subsequently, rectangularly positioned three circular patterns at three equally-separated focal depths along the z axis were photopatterned by the two-photon laser scanning confocal microscope. This patterning process was repeated for three times to yield rectangularly positioned three-dimensional structures of circularly patterned domains (a, a', a") (b, b', b") (c, c', c") (d, d', d").

Figure 2c depicts the confocal microscope image of the resulting three-dimensional structures. In Fig. 2c panel I the hydrogel pattern is imaged from the top side across the x-y plane. Four circular patterns of different brightness corresponding to a, b, c, and d are observed, consistent with the position of only pattern a in the imaging focal distance. Figure 2c panel II depicts the confocal image of the resulting pattern of the rectangularly positioned circular domains across the y-z plane at focal imaging distances corresponding to the pattern of the columns (a) and (c).

Three equally separated patterns along the z axis for column (a) and three separated columns (c) are observed, consistent with the 3D formation of (a, a', a") and (c, c', c"). Furthermore, for the circular patterns (c, c', c"), a shift in the z axis position is observed, consistent with the deeper laser patterning depth of the (c, c', c") domains as compared to the patterning depths of (a, a', a"). Figure 2c panel III depicts the confocal imaging of the resulting 3D pattern of the rectangularly positioned circular pattern consisting of the columns (b, b', b") and (d, d', d") upon focusing the confocal microscope to the x-z plane. The six separated patterns of (b, b', b") and (d, d', d") are observed and a clear position shift of the patterned columns (d, d', d") in respect to (b, b', b") is evident, consistent with the different lithographic focal depths. The results demonstrate the successful programmed three-dimensional patterning of the hydrogel film (Supplementary Movie 1 depicts the video of the computerized reconstructed 3D patterned hydrogel, demonstrating the patterns along the x-y-z axes). It should be noted that these lithographic patterning protocols lead to single strand tethers (2) on the non-irradiated domains. These tethers are available for subsequent spatially selective hybridization with any ligand, nanoparticle or biomolecule modified with the strand (2'). We note that in this photopatterning process the functionalization of the non-irradiated domains with fluorophore is demonstrated.

**Photopatterned activation of functional DNA hairpins within hydrogel films.** The opposite fabrication of a patterned hydrogel film that results in the activation of photolithographically patterned domain for the guided deposition of a fluorescent probe is presented in Fig. 3. In this system the hydrogel film is composed of the photocleavable hairpin monomer structure (3) and the non-photoresponsive acrylamide nucleic acid duplex module (1). The irradiation of this hydrogel film through the hole-containing

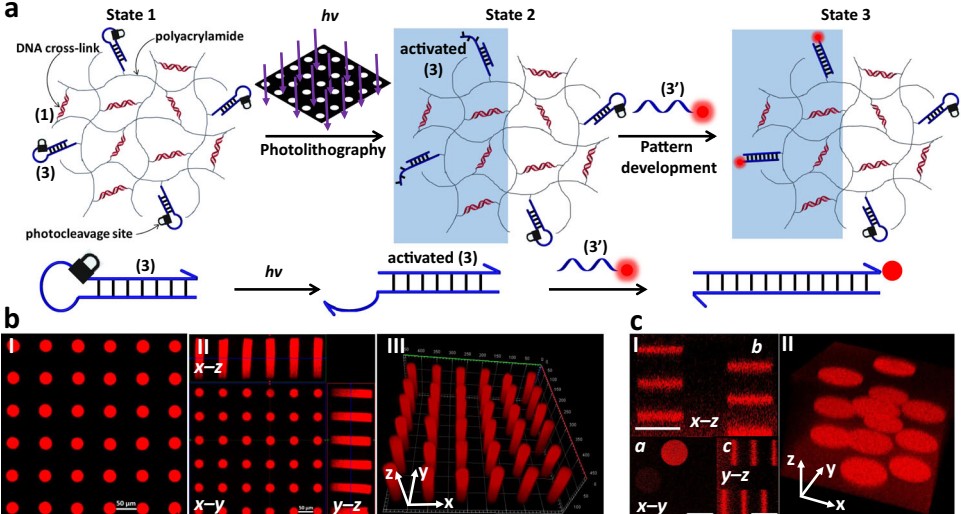

**Fig. 3 Light-induced patterning and imaging of a photoresponsive polyacrylamide hydrogel film functionalized with (1) and the o-nitrobenzylphosphate ester photocleavable hairpin (3). a** Photolithographic patterning of the hydrogel through a circular hole-containing mask ($\lambda = 365$ nm), and the fluorescence imaging of the resulting pattern by the hybridization of TAMRA labeled (3') to the activated (3) tethers associated with the irradiated domains. **b** Panel I—Fluorescence confocal microscope imaging of the patterned hydrogel. Panel II—Three-dimensional confocal microscope imaging of the hydrogel film across the x-y, x-z, and y-z planes. Panel III– Fluorescence confocal microscope imaging of the reconstructed three-dimensional patterned hydrogel. Scale bar lengths 50 μm. **c** Fluorescence confocal microscope imaging of twelve layered circular domains generated by two-photon laser scanning confocal microscope photopatterning process ($\lambda = 740$ nm) using different spatial and positional focusing depths. Panel I— Confocal microscope imaging of the two-photon laser patterned hydrogel fluorescence images recorded along the x-y plane (**a**), the x-z plane (**b**) and the y-z plane (**c**). Panel III– Confocal microscope imaging of reconstructed three-dimensional patterned hydrogel that includes twelve patterned discs in rectangular configurations along the x-y-z axes. Scale bars correspond to 20 μm. $N = 4$ independent experiments were performed.

mask leads to the photoactivation of the hairpin units in the light-exposed domains whereas the hairpin structures in the non-exposed areas are unaffected. The photoactivation of the hairpin units yields a toehold single-stranded tether associated with the fragmented duplex of the hairpin (activated (3) as shown in Fig. 3a). The guided hybridization of the TAMRA-functionalized nucleic acid (3') with the photolithographically generated pattern yields the fluorescent circles of TAMRA on the non-fluorescent background of the hydrogel film, Fig. 3b, panel I. The orthogonal fluorescence confocal microscope images of the patterned hydrogel along the x-z and x-y and y-z planes are shown in Fig. 3b, panel II, and the reconstructed three-dimensional patterned hydrogel is displayed in Fig. 3b, panel III. Clear three-dimensional red colored periodically separated cylinders across the hydrogel film are observed.

In addition, the maskless patterning of the hydrogel film using the two-photon laser scanning confocal microscope lithographic method ($\lambda = 740$ nm) was applied, Fig. 3c. As before the hydrogel film was photopatterned with twelve circular domains as described earlier along the x-y-z axes to yield the rectangularly spatially positioned circular patterns. The fluorescence confocal microscope images of the patterned domains are displayed in Fig. 3c. Figure 3c, panel I, shows the fluorescence images recorded along the x-y plane (a), the x-z plane (b) and the y-z plane (c). The images show the spatially separated patterns along the respective focal planes of the two-photon laser patterning process. Figure 3c, panel II, shows the reconstructed three-dimensional patterned hydrogel that includes twelve patterned discs in rectangular configurations along the x-y-z axes (Supplementary Movie 2 shows the video corresponding to the reconstructed 3D patterned hydrogel). It should be noted that this photopatterning protocols lead to a functional single-stranded tether in the photopatterned domains that enables the subsequent hybridization of any additional chemical component modified with a nucleic acid tether complementary to the fragmented hairpin

generated upon the light-induced patterning process. We note that the photopatterning process described in this section led to the functionalization of the irradiated domains with a free tether allowing secondary modification of these domains.

**Orthogonal photopatterning of hydrogel films exhibiting dual functionalities**. In the next system the composite hydrogel allowed the guided patterning and the deposition of the two fluorophores (fluorescein and TAMRA) on the hydrogel film, Fig. 4. In this system, the composite hydrogel consists of the duplex crosslinker nucleic acid module (1), the photocleavable module (2) and the photocleavable hairpin module (3). The irradiation of the hydrogel film through the hole-containing mask yields in the irradiated area the photodegraded hairpin duplex that includes the toehold tether, activated (3), as shown in Fig. 4a. In addition, the photocleavage of module (2) proceeds in the light exposed areas leading to the release of nucleic acid (2) from these domains. As a result, after the removal of the mask and washing of the hydrogel film, the light exposed patterned area is functionalized with the activated (3) for hybridization with TAMRA-modified (3') through toehold-mediated displacement reaction, whereas the non-irradiated area is functionalized with the single strand module (2) that leads to the hybridization with the complementary fluorescein-labeled strand (2'). As a result, the irradiation of the hydrogel shown in Fig. 4 leads to the selective association of TAMRA-(3') to the circular-separated patterns, and to the coverage of fluorescein-(2') on the background non-irradiated areas. Figure 4b, panel I shows the top image of the hydrogel composed of the fluorescent spatially-separated red circles of TAMRA on the green fluorescence background of fluorescein. Figure 4b, panel II shows the orthogonal side-imaging of the patterned hydrogel. The x-z and y-z confocal images show the inner deposition of TAMRA in the patterned circles. Figure 4b, panel III depicts the reconstructed three-dimensional

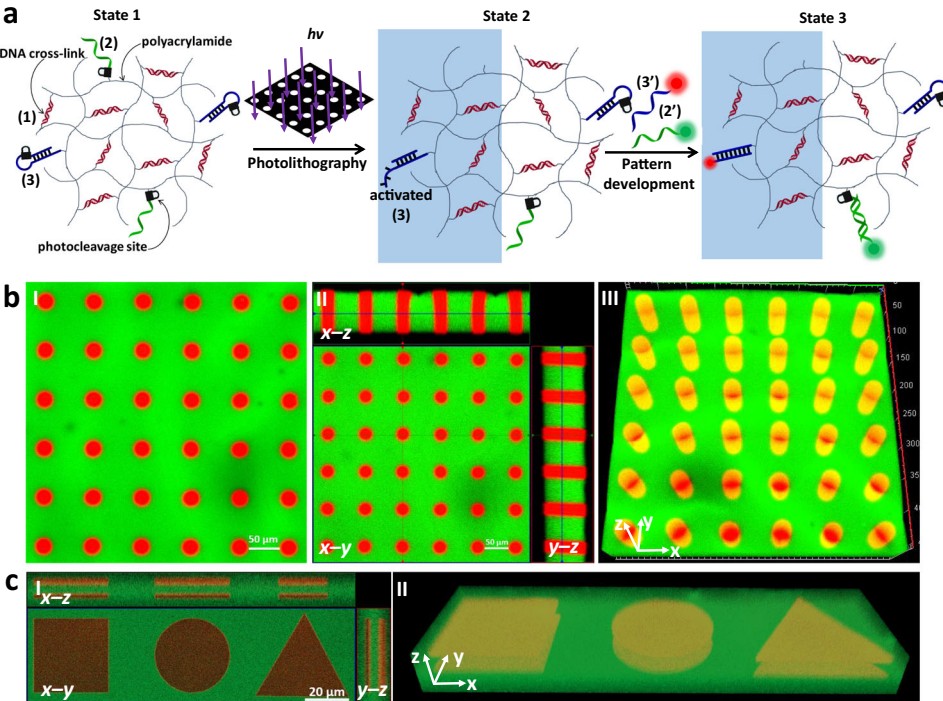

**Fig. 4 Light-induced orthogonal patterning of a hydrogel exhibiting dual functionalities. a** The polyacrylamide (1)-crosslinked hydrogel is functionalized with the o-nitrobenzylphosphate ester nucleic acids (2) and (3). Exposure of the hydrogel to light yields in the illuminated domains the activated strand (3) and retains in the non-irradiated domains the free nucleic acid strand (2). The dual patterned functionalities allow the selective association of (3')-TAMRA and (2')-FAM to the respective domains. **b** Panel I—Confocal fluorescence image of the circular periodic pattern of (3')-TAMRA on the continuous hydrogel functionalized with (2')-FAM. Panel II—Three-dimensional confocal fluorescence imaging of the patterned hydrogel according to the process outlined in Fig. 4a. Panel III—Reconstructed three-dimensional fluorescence image of the hydrogel film generated according to Fig. 4a. Scale bars correspond to 50 μm. **c** Panel I—Three-dimensional fluorescence imaging of two-layered rectangular, circular, and triangle structures patterned by scanning two-photon laser lithography. Panel II—Reconstructed three-dimensional pattern of the two-layered structures generated by the scanning two-photon laser lithography process. Scale bar corresponds to 20 μm. N = 4 independent experiments were performed.

photolithographically patterned hydrogel. Three-dimensional cylinders in a periodically spatially-separated order, consisting of TAMRA labeled hydrogel within the background, continuous, fluorescein labeled hydrogel are observed. Similarly, the photolithography process was applied to generate hydrogel film that includes ordered strip patterns, Supplementary Figure 1.

In addition, Fig. 4c panel I depicts the maskless photolithographic patterning of the hydrogel using two-photon laser scanning confocal microscope. In this experiment, square, circle, and triangle shapes were lithographically patterned in two focal layers of the hydrogel film. Figure 4c, panel I shows the confocal microscope images of the patterned domain along x-y, x-z, and y-z planes of the resulting pattern. Figure 4c, panel II depicts the reconstructed, maskless patterned, three-dimensional image of the generated two-layer structures using the two-photon laser scanning lithographic process. It should be noted that the light-induced patterning processes used in Fig. 4 yield hydrogel matrices modified by two different surface functionalities. The light exposed areas led to the three-dimensional functionalization of the hydrogel domains with the activated hairpin (3), while the domains that were not exposed to light include the tether (2) as functional ligand. These dual functionalities enabled the orthogonal modification of the respective patterns with (3')-TAMRA and (2')-FAM. In this section the photolithographic patterning led to different tethers functionalities linked to the irradiated and non-irradiated domains.

**Photolithographic patterning of periodically-separated non-crosslinked polymer circular domains within a continuous crosslinked hydrogel matrix.** Finally, Fig. 5 depicts the

photolithographic patterning of non-crosslinked circular polymer "hole" domains in a continuous crosslinked hydrogel matrix. The parent hydrogel film consists of the polyacrylamide matrix crosslinked by the photoprotected o-nitrobenzylphosphate ester functionalized duplex (4) to which the photoprotected nucleic acid tethers (2) and the photoprotected hairpins (3) are linked, Fig. 5a. Photoirradiation of the hydrogel through the circle-containing mask removes from the irradiated hydrogel areas the crosslinking units (4) and the single-strand tethers (2). Concomitantly, the photoprotected hairpin (3) is cleaved, resulting in the formation of the activated (3), in a form of a duplex consisting of a single-strand nucleic acid toehold. On the other hand, all background hydrogel domains, that were not exposed to the light-induced process, consist of the (4)-crosslinked hydrogel that is further functionalized with the photoprotected hairpin modules (3) and the single-strand photoprotected tethers (2), Fig. 5a. That is the patterned hydrogel film includes circular non-crosslinked polyacrylamide chains functionalized with the toehold-modified duplexes of the activated (3) units, surrounded by the (4) duplex crosslinked hydrogel modified with the single-strand tethers (2), and the caged hairpin units (3).

Supplementary Figure 2 shows the SEM images of the parent hydrated hydrogel film prior to the photopatterning process, panel I, and the photochemically patterned hydrogel film, panel II. The parent hydrogel film shows a porous structure consistent with a crosslinked hydrogel matrix. The photopatterning process leads to the apparent formation of spatially periodically separated "holes" on the background porous hydrogel surrounding. The results are consistent with the formation of non-crosslinked

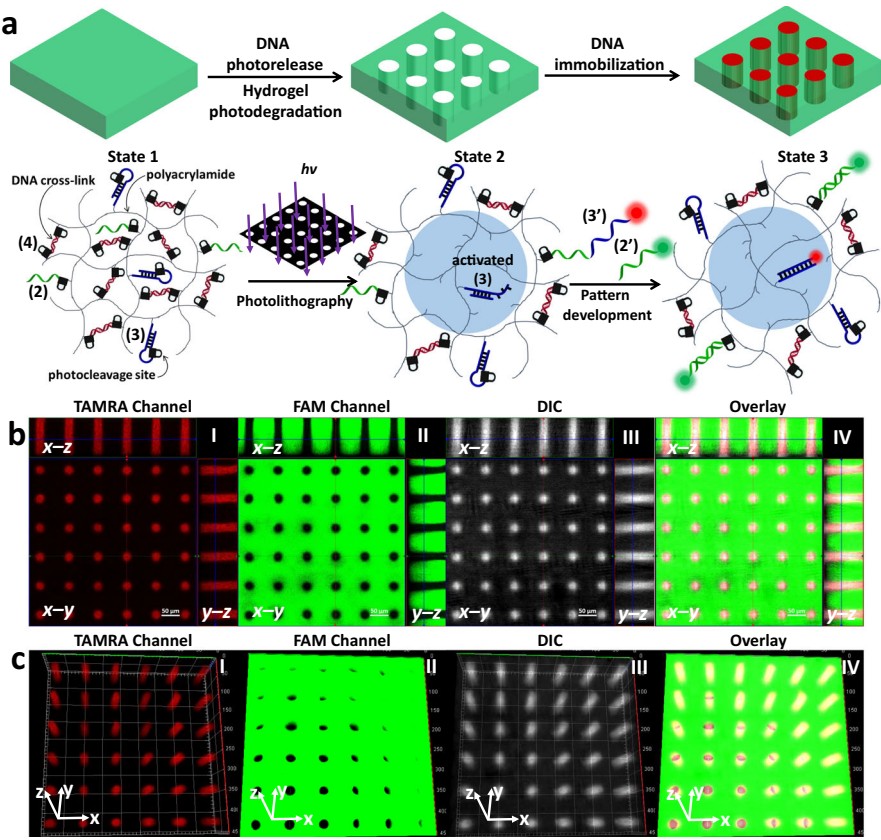

**Fig. 5 Photolithographic patterning of periodically-separated non-crosslinked polymer circular domains within a continuous crosslinked hydrogel. a** The patterned domains are functionalized with the activated (3) duplex in a continuous hydrogel film modified with the o-nitrobenzylphosphate ester modified units that include (2), (3) and the photocleavable crosslinker (4). The resulting photopatterned hydrogel film dictates the selective hybridization of TAMRA-modified (3') with the activated (3) units associated with the hole domains, and the hybridization of the FAM-functionalized (2') with the tethers (2) associated with the continuous hydrogel matrix. **b** Three-dimensional confocal microscope images of the photopatterned hydrogel film. Panel I—Confocal fluorescence microcopy image of the hydrogel film using the TAMRA channel. Panel II—Confocal fluorescence microscope image using the FAM channel. Panel III—Differential interference contrast (DIC) mode image. Panel IV—Overlay image of the patterned hydrogel film. Scale bars correspond to 50 μm. **c** Reconstructed three-dimensional confocal microscope images of the photopatterned hydrogel film. Panel I—The reconstructed TAMRA patterned hole domains. Panel II—The reconstructed image of the continuous FAM labeled hydrogel matrix. Panel III—The reconstructed image of patterned hole domains using differential interference contrast (DIC) mode image. Panel IV—The reconstructed overlay image of the patterned hydrogel film. N = 4 independent experiments were performed.

domains within the background of the intact hydrogel matrix. The resulting patterned hydrogel includes in the hole domains the functional tether of activated (3) linked to the non-crosslinked polyacrylamide modified holes, and the free tethers (2) associated with the continuous cross-linked hydrogel matrix. Subjecting the patterned hydrogel film to the TAMRA-modified strand (3') and the fluorescein-modified strand (2') results in the guided selective hybridization of (3') to the activated (3) tethers associated with the "hole" domains, and of the FAM-functionalized (2') to the (2) tethers linked to the background hydrogel film.

Figure 5b depicts the confocal microscopy images of the photopatterned hydrogel film across the x-y, x-z, and y-z planes using the TAMRA fluorescence channel, FAM-channel, differential interference contrast (DIC) mode and the overlayed image. The use of the selective TAMRA channel and FAM channel shows clearly the red fluorescence of the TAMRA patterned circles and the green fluorescence associated with the FAM functionalized hydrogel. The DIC image shows the bright-field image of the patterned interfaces where the difference between the stiffness of the non-crosslinked circle domains and the higher stiffness of the continuous crosslinked hydrogel enables the imaging of the patterned interface. The reconstruction of the confocal microscopy images is depicted in Fig. 5c where the

reconstructed TAMRA patterned circular domains, the continuous FAM labeled matrix, the DIC patterns and the resulting overlay image are presented. The reconstructed TAMRA labeled image, the DIC image, and particularly the overlay image provide the effective optical profile of the interior of the hole structure, demonstrating the pattern formation across the entire width of the hydrated hydrogel film. (For a further visualization of the optical profiling of the holes see Supplementary Movie 3).

This patterning process leads to holes consisting of circular domains of low degrees of crosslinking (presumably of very low stiffness) surrounded by a continuous hydrogel exhibiting enhanced crosslinking (presumably of higher stiffness). These features are supported by rheology experiments, Supplementary Fig. 3. The parent hydrogel reveals G' ≈ 1170 Pa, G" ≈ 220 Pa whereas the UV irradiated hydrogel reveals a quasi-liquid low stiffness behavior, G' ≈ 460 Pa, G" ≈ 320 Pa. In this section the photopatterning procedure led to the formation of spatially periodically separated holes in a continuous hydrogel surrounding.

**DNA hybridization chain reaction (HCR) dictates site-specific swelling of the patterned hydrogel film for pattern modulation.** The photopatterning procedures presented in Figs. 2–4

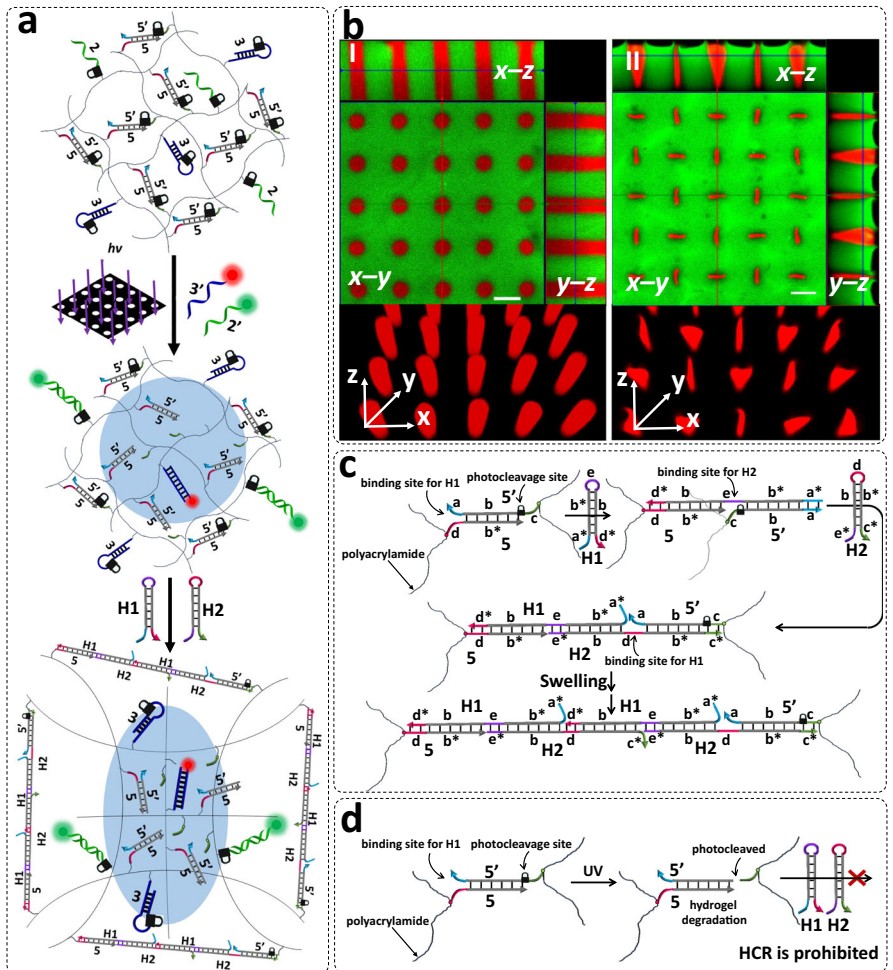

**Fig. 6 DNA hybridization chain reaction (HCR) dictates site-specific swelling of hydrogel films for pattern modulation. a** Photolithographic patterning of a o-nitrobenzylphosphate ester photoprotected hydrogel that yields non-crosslinked periodically spaced holes on a continuous hydrogel film crosslinked by photoprotected (5)/(5') duplexes and functionalized with (2) and hairpin (3). The photolithographic process leads to the guided hybridization of the TAMRA-(3') and FAM-(2') with activated (3) and (2), associated with the hole domains and the continuous hydrogel, respectively. The intact crosslinker units bridging the continuous hydrogel film initiate the hybridization chain reaction (HCR), in the presence of hairpins H1 and H2 to yield long-chain nucleic acid wires bridging the polyacrylamide scaffold. **b** Panel I—Three-dimensional confocal fluorescence microscope images and reconstructed image (bottom panel I) of the spatially-separated periodic circular (cylindrical) patterned structures generated according to Fig. 6a. Panel II—Three-dimensional confocal microcopy image of the spatially-separated orthogonal ellipsoid microstructures and the reconstructed structures (bottom panel II) generated upon the forced transition of the circular patterned domains into the ellipsoid structures as a result of the HCR process proceeding in the continuous hydrogel film according to Fig. 6c. Scale bars correspond to 50 μm. **c** Schematic mechanism of the hybridization chain reaction (HCR) initiated by the duplexes (5)/(5') in the presence of hairpins H1 and H2, leading to long DNA duplex wires bridging the polyacrylamide network. **d** Schematic inhibition of the HCR process by the photodeprotected duplexes (5)/(5') positioned in the "hole" domains. $N = 4$ independent experiments were performed.

demonstrated the selective functionalization of the patterned domains with tethers that enabled secondary dictated hybridization of target nucleic acid to the respective patterned domains. In the last photopatterning process, Fig. 5, we introduced a method to pattern the hydrogel film with two domains consisting of periodically-separated non-crosslinked, circular polymeric domains, positioned in a continuous crosslinked hydrogel matrix. Such photopatterned hydrogel would allow us to examine the effect of chemically-induced strain changes within the crosslinked hydrogel matrix on the spatially-separated, lower-stiffness, non-crosslinked circular domains. Towards this goal we designed the photoresponsive hydrogel presented in Fig. 6.

In the first step, Fig. 6a, a hydrogel film consisting of the polyacrylamide hydrogel crosslinked by the photoprotected duplexes (5)/(5') (where the strand (5') is functionalized with the o-nitrobenzylphosphate ester protective group) and further modified with the photoprotected stand (2) and the

photoprotected caged hairpin (3). The photochemical irradiation of the hydrogel film through the hole-containing mask cleaves the crosslinking units (5)/(5'), and concomitantly cleaves-off the tethers (2), and uncages the hairpin units (3) to yield the activated (3) units. This patterning procedure yields circular patterns consisting of non-crosslinked, low stiffness, domains decorated with the toehold modified activated (3) units, surrounded by a (5)/(5') crosslinked hydrogel functionalized with the free tethers (2) and the caged hairpins (3). Subjecting the resulting patterned hydrogel to the FAM-modified (2') and to the TAMRA-functionalized (3') leads to the selective spatial hybridization of the fluorophore labeled strands to the matrix.

The confocal microscope image of the resulting patterned hydrogel film is depicted in Fig. 6b, panel I. The x-y image of the hydrogel reveals equally-spaced circular domains (red fluorescence, TAMRA), where the green background corresponds to the FAM-(2')-modified hydrogel. The orthogonal x-y, x-z and y-z

fluorescence images and the reconstructed patterned structures (bottom of panel I) demonstrate that the irradiated domains are composed of circular three-dimensional cylinders. In the next step, the patterned hydrogel film was subjected to the two hairpins H1 and H2. The crosslinking units (5)/(5') compositing the continuous hydrogel film act as promoters for the initiation of the hybridization chain reaction (HCR), as schematically presented in Fig. 6c. That is, the crosslinking unit (5)/(5') were extended by H1 and H2 through HCR process, forming (5)-H1-H2-H1-H2…-(5') long negatively charged DNA polymer chains. Accordingly, the HCR process is anticipated to yield long negatively charged DNA polymer chains that crosslink the polyacrylamide chains. It should be noted, however, that the photocleavage of the (5)/(5') crosslinking bridges in the hole domains lack the appropriate initiator strands (cf. Fig. 6a, d) to promote the hybridization chain reaction. Thus, the HCR process is prohibited in the circularly patterned domains. The resulting hydrophilic HCR-functionalized hydrogel film is then anticipated to swell, and is expected to exert a strain force on the periodically spatially-separated non-crosslinked hole patterned domains associated with the film.

Figure 6b, panel II depicts the confocal microscope image of the resulting patterned surface subjected to the HCR process. The hydrophilicity and accompanying swelling of the continuous background crosslinked hydrogel film have significant consequences on the shape of the periodically-spaced non-crosslinked irradiated domains. The x-y fluorescence images show that the circular domains are narrowed into ellipsoid shaped patterns. Moreover, the ellipsoid shapes reveal an ordered orthogonal alternating shape transition, where each horizontal ellipsoid is surrounded by four perpendicular ellipsoids separated by 90°, and vice versa. The orthogonal confocal microscope images and the resulting reconstructed three-dimensional structures of the HCR generated hydrogel reveal that the strain forces exerted on the low stiffness symmetric cylinders have a significant effect on the three-dimensional structure of the photolithographically patterned domains, Fig. 6b, panel II and Supplementary Fig. 4.

Furthermore, the formation of orthogonal alternating ellipsoid shapes is supported by a SEM image of the HCR-generated pattern, Supplementary Fig. 5 that shows dense areas between pores. To account for these regular pattern transitions created on the non-crosslinked spatially-separated circular domains, by the bulk HCR-stimulated swelling of the surrounding hydrogel matrix, we refer to experimental and computational studies addressing pattern transformations of periodic soft material structures triggered by deformation[78]. Experimental and computational-simulated experiments[79] demonstrated that subjecting a soft polymer sheet containing periodically spaced holes to a macroscopic strain of $\varepsilon = 0.10$ led to the collective ordered deformation of the hole structures, Supplementary Figure 6. The pattern of circular holes underwent compression and the transformation of the holes into mutually orthogonal ellipses. These pattern transitions were attributed to the compressive loading of the vertical interhole ligaments, which undergo a buckling instability, forcing the system to the new configuration. Presumably, this mechanism is, also, operative in the pattern transitions observed in the photopatterned DNA hydrogel, Fig. 6b.

These results extend previous impressive reports on photolithographic patterning of DNA hydrogels through light controlled alternation of the stiffness of the matrices via controlling the crosslinking degrees[80] or by inducing stiffness changes through the hybridization chain reactions[66]. The significant advance is the relation between the HCR induced stress in the bulk hydrogel matrix on ordered spatially separated low-stiffness, circular domains. That is, the stress generated in the bulk matrix

results in ordered orthogonal shape transitions of the circular domains into orthogonal alternate ellipsoidal structures. In this section the effect of the stress induced by the hybridization chain reaction (HCR) proceeding on the continuous hydrogel area on the shapes of the low-stiffness circular domains is discussed. The transition of the circular domains into ordered orthogonal alternate ellipsoidal structures is demonstrated.

**Selective binding and proliferation of HeLa cells in the confined MUC-1 aptamer patterned domains.** Finally, the photochemical patterning of the o-nitrobenzylphosphate ester photoprotected hydrogels described in Figs. 5–6 was used for the spatial assembly of HeLa cancer cells. Specifically, the photo-patterning of spatially periodically separated non-crosslinked cylindrical domains within the continuous crosslinked hydrogel was applied to deposit the cells in the non-crosslinked reservoirs. The hydrogel film prior to the photopatterning was composed of the polyacrylamide polymer functionalized with the protected hairpins (3) and the protected crosslinking duplexes (4), Fig. 7a. The light induced patterning of the hydrogel film through a circular hole-containing mask yielded non-crosslinked cylindrical polymer domains, modified by the uncaged hairpin consisting of the toehold-functionalized duplex activated (3), embedded in the continuous (4)-crosslinked hydrogel that was modified with the caged hairpin (3), Fig. 7a. The hole-containing domain of the patterned film were further modified by the hybridization of TAMRA-modified strand (3") that includes the MUC-1 aptamer conjugated to a strand domain complementary to (3), with the toehold tether of activated (3). The mucin 1 protein (MUC-1) is an overexpressed glycoprotein associated with HeLa cells surface membranes. The anti-MUC-1 selected DNA aptamer[81] reveals specific high-affinity binding toward MUC-1,[82] interactions that were previously applied for imaging and sensing of HeLa cells.[74,82] The MUC-1 aptamers recognize HeLa cells, thus, providing functional tethers for the selective capturing of HeLa cells in the confined patterned domains. Subsequently, subjecting the captured cells on the patterned domains to the cell growth medium allowed the probing of cell proliferation on the target patterned microstructures.

Figure 7b, panel I depicts the confocal fluorescence image of the TAMRA-(3") associated with the patterned domains. Panel II shows the bright-field image of the patterned hydrogel film after treatment with the HeLa cells for a time-interval of two hours. The selective MUC-1 aptamer dictated adsorption of the HeLa cells on the patterned domains is visible. Panel III shows the overlay images of panel I and panel II, demonstrating the co-adsorption of TAMRA-(3") and the HeLa cell on the patterned domains. Similarly, MUC-1 aptamer patterned hydrogels with different patterns, such as strip-like and hexagon shapes were fabricated, and the HeLa cells were selectively adsorbed onto the patterned domains as shown in Supplementary Figs. 7 and 8. Figure 7c and Supplementary Fig. 9 show the respective confocal microscope images of the TAMRA-(3") and HeLa cells after incubating the cell-modified patterned film in the cell growth medium for a time-interval of 24 h. Selective cell proliferation in the patterned domains is visible reflected by the formation of aggregated cells. Thus, the cell proliferation is confined to the patterned domains.

In addition, complementary rheology experiments examined the effects of the proliferated cells confined to the patterned domains on the stiffness of the supporting hydrogel matrix, Supplementary Fig. 10. In this experiment the stiffness of the bulk hydrogel comprising the circular domain of the hydrogel generated by UV irradiation of the film shown in Fig. 7 was examined before the cell deposition, G' ≈ 350 Pa, G" ≈ 220 Pa, and

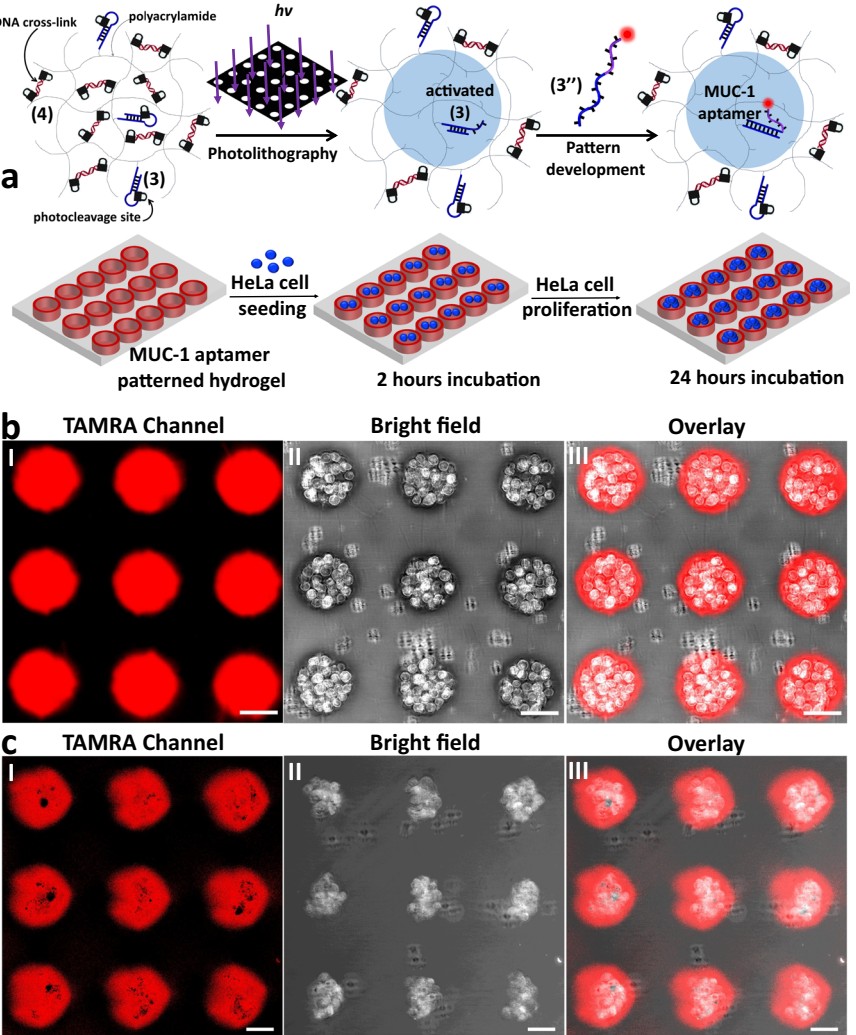

**Fig. 7 Patterning of a photoresponsive hydrogel for the selective binding and proliferation of HeLa cells in the confined patterned domains. a** Photopatterning of circular domains functionalized with the toehold activated duplexes in the continuous (4)-crosslinked hydrogel matrix modified with hairpins (3). The TAMRA-(3″) conjugated with MUC-1 aptamer are hybridized with the activated (3)-duplex to yield functional domains for the selective binging of HeLa cells and for the concomitant fluorescence imaging of the patterned cell structures. **b** Panel I—The confocal fluorescence microscope image of the TAMRA-(3″) modified patterned hydrogel film. Panel II– Bright field image of the HeLa cells linked to the confined circular domains modified with MUC-1 aptamer. Panel III– Overlay image of panel I and panel II. Scale bars are 50 μm. **c** Panel I—Fluorescence confocal microcopy image (TAMRA channel) of the TAMRA-(3″) in the patterned domains. Panel II—Bright field confocal image of the proliferated cells in the patterned domains. Panel III—Overlay of panel I and panel II. Scale bars are 50 μm. $N = 4$ independent experiments were performed.

after the cell deposition and proliferation, $G' \approx 110$ Pa, $G'' \approx 80$ Pa. The result demonstrates that the polymer support that includes the proliferated cells is composed of a very soft, quasi-liquid phase, suggesting that, the stress-induced forces of the surrounding hydrogel can, indeed, dynamically deform the shapes of the cell aggregates. Furthermore, the viability of the proliferated cells in the patterned domains should be addressed. For this, the proliferated cells in the patterned domains, after a time interval of 24 h, were stained with calcein-AM and propidium iodide (PI) to identify the living cells and dead cells, respectively, and the patterns were imaged by the respective fluorescent channels, Supplementary Fig. 11. The results demonstrate that only the fluorescent patterns of the calcein-AM stained domains are observed, Supplementary Fig. 11, panel I, whereas no visible fluorescence of dead PI-stained cell is detectable, Supplementary Fig. 11, panel II, thus no detected cell motility could be realized after the imaging time interval (for complementary bright-field and overlay images of the stained domain see Supplementary Fig. 11, panel III and IV, respectively).

**Selective proliferation of HeLa cells in confined MUC-1 aptamer functionalized ellipsoid microstructures.** The successful transition of periodically spaced circularly-separated non-cross-linked patterned domains into alternate orthogonal ellipsoids shapes, where each horizontal ellipsoid is surrounded by perpendicular four ellipsoid separated by 90°, and vice versa. Compressive stress forces exerted by the HCR process proceeding in the surrounding hydrogel, was further applied to grow programmed orthogonal ellipsoid structures of HeLa cells, Fig. 8. In this experiment, the circular patterned domains prepared according to Fig. 6a, labeled with the TAMRA-(3″), were seeded with the HeLa cells, and allowed to proliferate while subjecting the continuous hydrogel to the HCR process for a time interval of ten hours, according to Fig. 6c. The cultured cells were then stained with calcein-AM to indicate the position and vitality. The cell cultures underwent a stress induced growth within the confined orthogonal ellipsoids structures. Figure 8, panel I shows the fluorescence image of the calcein-AM stained HeLa cells proliferated in the ellipsoid microstructures, panel II shows the

| Calcein-AM Channel | TAMRA Channel | Overlay |
| --- | --- | --- |

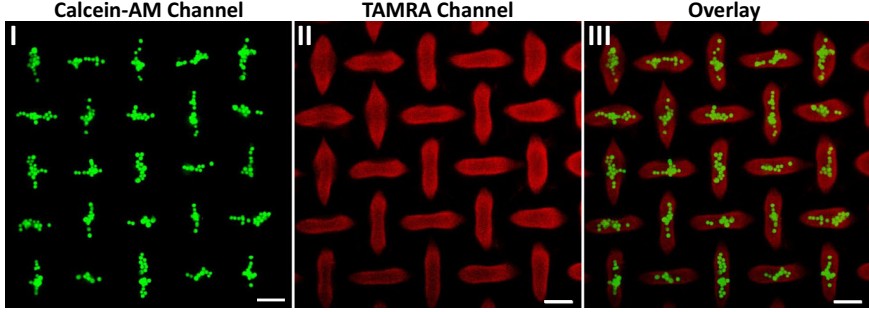

**Fig. 8 Confocal microscope images of HeLa cells bound and proliferated in confined ellipsoid microstructures formed upon the HCR-induced process in the continuous hydrogel film.** Panel I—Fluorescence microcopy image (Calcein-AM channel) of the calcein-AM stained HeLa cells proliferated in the ellipsoid microstructures. Panel II—Fluorescence microcopy image (TAMRA channel) of the ellipsoid microstructures modified with MUC-1 aptamer. Panel III—Overlay of panel I and panel II. Scale bars correspond to 50 μm. $N = 3$ independent experiments were performed.

TAMRA-(3") in the ellipsoid microstructures and panel III shows the overlay image. Alternating, periodically-separated orthogonal ellipsoid aggregated cell shapes are formed and all of the HeLa cells were confined in the ellipsoid microstructures. Each horizontal ellipsoid aggregated cell structure is surrounded by four perpendicular ellipsoid aggregated cell structures, separated by 90°, and vice versa. Thus, the results demonstrate a possible future path to generate programmed intercommunicating cell microstructures.

In conclusion, the study has introduced a versatile method that applies combinations of o-nitrobenzylphosphate photoprotected hydrogel films to yield a universal set of photopatterned films exhibiting guided functionalities and diversified control over the stiffnesses of the patterned domains. Two methods to photopattern the matrices were discussed: One method has involved the microscale photolithographic patterning of the hydrogel films by irradiation of the matrices through a mask (a hole-containing mask with periodic circular patterns), $\lambda = 365$ nm. The second approach has demonstrated the use of two-photon laser scanning confocal microscope patterning, of three-dimensional microstructures, in the hydrogel films ($\lambda = 740$ nm). By altering the composition of the photoresponsive hydrogel films, and applying the two photopatterning methods, the formation of different spatially-structured hydrogel films mimicking extracellular environments composed of variable topographic, and spatially-dictated ligand modifications, for secondary targeted attachment of chemical agents, were demonstrated. These unique properties of the patterned hydrogels were successfully applied for the aptamer-guided deposition and proliferation of HeLa cells in the photopatterned hole-containing domains of the spatially confined microcontainers.

An important result of the present study has involved the control over the shapes of periodically separated circular non-crosslinked polymer domains within a continuous hydrogel film in which strain deformations prevail. That is the strain in the continuous hydrogel environment resulted in ordered compressive shape transitions of the periodic circular domains into ordered orthogonal alternating ellipsoid structures, where each horizontal ellipsoid is surrounded by four perpendicular ellipsoids separated by 90°, and vice versa. Specifically, the hybridization chain reaction (HCR) initiated by promoter nucleic acid tethers patterned on the continuous hydrogel film led to swelling of the film and to an anisotropic stress, leading to the ordered deformation of the circular domains. Beyond the stress-induced topographic transitions of the circular domains into ordered ellipsoid structures, the guided deposition of HeLa cells into the circular domains allowed the HCR-stress dictated proliferation of the HeLa cells in the secondary structured ellipsoid containments.

It should be noted that different parameters, such as the distance separating the periodic patterns, the size of the patterns and the stiffness of the surrounding hydrogel matrix are anticipated to control the stress-induced shapes of the patterned domains[78,79], and thus regulate the shape of the cell aggregates proliferated in the patterned domains. As stated, the stress-induced topographic transitions of the circular domains into ordered ellipsoid structures are anticipated to be controlled by the size of the circular domains, the spatial separation distance of the periodic domains and the stiffness of the bulk hydrogel matrix surrounding the circular patterns[78,79]. Accordingly, the shapes of the proliferated cells and the inter-pattern communication of cells are expected to be regulated by the auxiliary topographies of the patterns. Our study raises important challenges that need to be resolved, such as the characterization of the dynamics of cell proliferation in the patterned domains on the evaluation of the effects of domains sizes and mechano-effects of the surrounding matrix on the cell growth. These issues, far beyond the scope of the present study, pave, however, future challenges to examine neighboring cell-cell interactions at microscale separation distances. In addition, it would be meaningful to reversibly switch the stiffness of the surrounding hydrogel matrix by auxiliary triggers thereby controlling the reversible switching of the patterned topologies and the shapes of the proliferated cell structures. In the present study, we used an irreversible HCR-guided formation of the stiff hydrogel (the reverse separation of the HCR matrix by a strand-displacement process is very inefficient). Nonetheless, recent studies demonstrated the HCR-guided synthesis of stimuli-responsive hydrogels revealing switchable stiffness properties using triggers, such as $K^+$ ion/ crown ether[64] or pH-responsive T-A·T triplex[83] units. Adapting these principles to switch the stiffness of the surrounding hydrogel matrices could lead to the switchable shapes of the proliferated cells.

## Methods

**Materials.** Ultrapure water with 18.2 mΩ•cm was used in all experiments (Heal Force water purification system). All chemical reagents were analytical grade and used without further purification. All DNA oligonucleotides used were purchased from Sangon Biotechnology Co., Ltd. (Shanghai, China) and Hippo Biotechnology Co., Ltd. (Beijing, China) and purified by HPLC. The DNA sequences are shown in Supplementary Table 1-Supplementary Table 4. HeLa cells used in all experiments were obtained from ATCC (ATCC Number: CCL-2). 3-(Trimethoxysilyl) propyl methacrylate was purchased from Sigma-Aldrich Co., Ltd. Acrylamide, ammonium persulfate (APS) and TEMED were purchased from Bio-Rad Co., Ltd. Calcein-AM and propidium iodide (PI) were obtained from Yeasen Biotech Co., Ltd. (Shanghai, China). The buffer used in all experiments was TE buffer (20 mM Tris-HCl, 150 mM NaCl, and 5 mM MgCl₂; pH 7.4).

**Hydrogels chamber preparation.** The chamber is composed of two glass slides. A spacer (polyimide tape) was mounted on a fluorine resin coated glass slide and the

respective pre-gel mixture was placed within the spacer domain and covered with a counter silanized glass slide. The top silanized glass slide served as a substrate onto which the hydrogel samples adhered after polymerization. The bottom fluorine resin coated glass slide prevented the DNA gel from sticking to the glass surface and allowed for minimal edge roughness. For the preparation of fluorine resin coated glass slide, the glass slide was treated with $O_2$ plasma for 1 min to make the glass surface hydrophilic. The treated glass slide was immersed in a solution containing fluorine resin solution (Item NO: SUREC 2101 S, donated by AGC Chemicals) at room temperature, followed by lifting at a rate of 16 mm/s. Next, the glass slide was dried at room temperature and baked at 120 °C for 30 min. For the preparation of silanized glass slide, the glass slide was sonicated in 10% (w/w) NaOH solution for 30 min, rinsed with MilliQ water, and dried under $N_2$ gas. The cleaned glass slide was, then, incubated in 4% (v/v) 3-(Trimethoxysilyl) propyl methacrylate isopropanol solution with 0.2% (v/v) acetic acid for 2 h. The resulting modified glass slide was rinsed with absolute isopropanol, and dried with $N_2$ gas.

**Preparation of pre-gel solution**. The gel was prepared by free radical copolymerization of acrylamide along with a DNA duplex as the crosslinker. To enable their copolymerization with acrylamide, all DNA strands were modified at the 5'-end with an acrydite unit. The pre-gel solution was prepared by mixing 40% (w/w) acrylamide, initiator (10 mg APS was dissolved in 95 µL TAE buffer with 5 µL TEMED), crosslinker DNA, photocleavable single-stranded DNA and photocleavable hairpin DNA. The final concentrations of all pre-gel components are as follows: 1 mM crosslinker DNA (1 or 4), 10% (w/w) acrylamide, 200 µM photocleavable single-stranded DNA (2) or 200 µM photocleavable hairpin DNA (3) and 0.11 M APS.

**Formation of photoresponsive DNA-based hydrogels**. Gels were formed by adding pre-gel solution into the hydrogel chamber for free radical polymerization. Thus, DNA hydrogels with controllable thickness were prepared on the silanized glass slide surface. After gel formation, the gel-coated glass slide was transferred into the TE buffer and incubated in TE buffer overnight to remove the fluorine resin coated glass. In this way, the hydrogels with desirable thickness were fabricated on the 3-(Trimethoxysilyl) propyl methacrylate silanized glass surface.

**Photopatterning process**. Photomask based patterning was performed through programmable UV irradiation using designed photomask with desired patterns. The hydrogel film was irradiated by UV light (365 nm, 15 mW/cm$^2$) for 5 min through photomask. All the two-photon lithography procedures were performed by using LSM 880 confocal microscope (Carl Zeiss) equipped with a FemtoSecond Laser (Coherent Inc.). A 40 × objective and 740 nm FemtoSecond Laser with 8% power were used in whole lithography process. After irradiation, the staining solution, 100 µM (2') or (3'), was added to develop the patterns. The patterned hydrogels were observed using confocal microscope (Carl Zeiss, LSM 880).

**Scanning electron microscope imaging**. SEM images were taken with Extra High-Resolution Scanning Electron Microscope (Hitachi SU8010) at 10 kV. The patterned hydrogels on glass surface was frozen by immersing it in liquid nitrogen. The frozen sample was dried by sublimation of the formed ice under high vacuum, and further metal-coated with Au.

**Site-specific swelling of the patterned hydrogels**. For the hydrogel pattern modulation via site-specific DNA HCR-directed programmable expansion, poly-acrylamide/DNA hydrogel was prepared by using pre-gel solution containing 500 µM strand (5), 500 µM strand (5'), 200 µM strand (2), 200 µM strand (3) and 10% (w/w) acrylamide. After gel formation, the hydrogel was photopatterned and imaged using confocal microscope. Subsequently, 100 µM H1 and H2 mixture was added to the patterned hydrogel and incubated overnight to perform DNA HCR-directed programmable expansion process and the swollen hydrogel was imaged using confocal microscope under 25 °C.

**Cell culture on MUC-1 aptamer patterned hydrogel surface**. For the cell culture experiment, MUC-1 aptamer patterned hydrogels was prepared according to the following procedure. The hydrogel was prepared using pre-gel solution containing 1 mM strand (4), 200 µM strand (3) and 20% (w/w) acrylamide. After gel formation, the hydrogel was photopatterned and modified with the MUC-1 aptamer tailed strand (3"). Subsequently, about 1×10$^7$ HeLa cells (obtained from ATCC) in 1 mL cell culture medium were seeded on the patterned surface. It should be noted that in contrast to the different patterned hydrogels in the study 10% w/w acrylamide hydrogel, we observed that this composition of the hydrogel is unstable in the presence of the cell-medium growth. Accordingly, for the cell-patterned hydrogels we applied a 20% w/w acrylamide hydrogel. The cells were further cultured in 1640 medium (Gibco) supplemented with 10% fetal bovine serum (Gibco) and 0.5 mg mL$^{-1}$ penicillin-streptomycin at 37 °C, using a humidified 5% $CO_2$ incubator. The cultured HeLa cells were stained with 4 µM calcein-AM and 4 µM propidium iodide (PI) for 30 min to verify the viability of the cells. Cultured cells were observed at different time intervals using confocal microscope.

**Cell culture on MUC-1 aptamer patterned orthogonal ellipsoid structures**. The hydrogel film was prepared by using pre-gel solution containing 500 µM strand (5), 500 µM strand (5'), 200 µM strand (3) and 20% (w/w) acrylamide. After gel formation, the hydrogel film was photopatterned using MUC-1 aptamer tailed strand (3") to obtain the circular patterned domains. The patterned hydrogel film was seeded with the HeLa cells, and allowed to cell proliferate while subjecting the continuous hydrogel to the HCR process with the addition of H1 and H2 into the cell culture medium for a time interval of ten hours. Cultured cells were imaged at different time intervals using confocal microscope.

**Rheological measurements**. Rheological properties were determined by a DHR-2 rheometer (TA Instruments) equipped with a 20 mm parallel plate geometry with a gap size of 0.2 mm. The experimental temperature was fixed at 37 °C by a cyclic water bath. The time sweep measurements were performed at a 1% strain with a fixed frequency of 1 Hz.

**Statistics and reproducibility**. It should be noted that for the photopatterned images shown for the different systems, no noticeable difference between $N = 3–4$ experiments could be identified.

**Reporting summary**. Further information on research design is available in the Nature Research Reporting Summary linked to this article.

## Data availability
All relevant data that support the findings are available within this article and Supporting Information and are also available from authors upon reasonable request. Source data are provided with this paper. A reporting summary for this Article is available as a Supplementary Information file. Source data are provided with this paper.

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

## Acknowledgements

We thank Prof. Daniel Harries, Institute of Chemistry, The Hebrew University of Jerusalem for the helpful scientific discussions. We thank prof. Weiwei Guo, Research Centre for Analytical Sciences, Tianjin Key Laboratory of Biosensing and Molecular Recognition, Nankai University for the helpful rheological measurements. This work is supported by the National Natural Science Foundation of China (21974127, 22090050, 21874121), the National Key Research and Development Program of China (2018YFE0206900), Hubei Provincial Natural Science Foundation of China (2020CFA037), Zhejiang Provincial Natural Science Foundation of China under Grant No. LD21B050001 and Grant No. LY20B050001.

## Author contributions

F.H., F.X. and I.W. directed and conceived the project. F.H. designed and performed experiments and carried out data analysis with help from M.C., Z.Z., and R.D. F.H., F.X. and I. W. co-wrote the paper. All authors discussed the results and commented on the manuscript.

## Competing interests

The authors declare no competing interests.
