## [Peer Review File · Nature Communications]

Reviewers' Comments:

Reviewer #1:

Remarks to the Author:

In their manuscript "Spatiotemporal Patterning of Photoresponsive DNA-Based Hydrogels Functional Matrices for Dictated Growth of Cells", Huang et al. prepared a series of o-nitrobenzylphosphate ester nucleic acid-based polyacrylamide hydrogel films as photoresponsive matrices for dictated growth of cells. The authors choose two methods, namely photolithographic patterning and 2-photon laser scanning confocal microscopy, to photopattern the hydrogel. In this way, the authors generated various spatially structured hydrogel films with controlled stiffness, hydrophilicity and ligand substitution. Notably, they were able to prepare ordered orthogonal alternate ellipsoidal structures through ordered transition from circular holes. The prepared structures were applied to dictate 3D growth of cells in a confined nanoenvironment. This is an interesting study which shows promising results for the dictation of 3D spatial cell growth. The study is well-executed and the manuscript is well written. It will be of interest to a broad scientific community. The manuscript can be considered for publication after the authors addressed the following concerns:

1. The photocleavable linkage adopted here is similar to the one reported previously by the authors (Nano Lett. 2019, 19, 1, 618–625) and should be cited.
2. The manuscript is entitled 'spatiotemporal patterning hydrogels ... for dictated growth of cells'. The term "dictation" could also indicate that the spatiotemporal variation of structure affect the cells behavior (e.g., dictation, migration, proliferation, etc.), which is not shown in the manuscript. Perhaps "dictated 3D growth" may be more suitable.
3. Will it be possible to comment on the maximum depth profile that can be achieved with the photopatterning?
4. Fig. 5: Can the pattern development into orthogonally arranged ellipses also be shown via SEM? Could denser areas between pores be created?
5. The 3D growth of cells in the photopatterned gel is convincing but the authors should also prove biocompatibility (e.g., live/dead cell staining). Live cell staining will also be valuable to provide the 3D distribution of living cells inside the holes.
6. Please provide the statistical analysis and error bars. Also include all experimental details (e.g., cell concentrations used for seeding and adsorption to the patterned areas). Why was 20% (w/w) acrylamide gels when performing the cell assays but 10% (w/w) for the rest of the studies?
7. Minor issues:

Is there any effect of the periodicity of the patterning on cell growth? It will be valuable if the authors can comment on this aspect.

Fig. 4: It will be valuable if the authors can comment whether the signals be switched off again e.g. by competitive strands (also shown in earlier works). In addition, what is the minimum distance between the ellipses?

p.13 l.303, red comma

Data on 'programmed intercommunicating' is missing (page 24, line 549-551).

Reviewer #2:

Remarks to the Author:

The manuscript by Huang et al. highlights a clever use of o-nitrobenzylphosphate esters to generate photoresponsive DNA-based hydrogels. The chemistry is nicely executed and the materials are described well. The results and possibilities are interesting, especially the use for the selective adhesion of cells and generation of anisotropic patterns through secondary chain growth using the hybridization chain reaction. However, there are several aspects of the manuscript that should be improved prior to publication.

1. In the Abstract / Introduction, the specific contribution of this work is not made clear. There are general discussions about photoresponsive materials (many of which already exist) and the

specific focus of this work is only evident later in the text. The authors should help guide the reader to the unique aspects of this work and better frame it within the context of the broader field.

2. In this context, there are several research topics from the field of photoresponsive hydrogels (for biofunctionalization or cell culture) that should be acknowledged. There are much broader use of nitrobenzyl compounds in the design of photoresponsive gels. For example, they have been used to generate photo tunable acrylamide gels [Frey and Wang *Soft Matter*, 2009, 5, 1918]; substrates for sub cellular detachment with two-photon exposure [Tibbitt et al. *Soft Matter* 2010, 6, 5100]; substrates for capture and release of cells [LeValley et al. *Colloids and Surfaces B: Biointerfaces* 2019, 174, 483]; patterning of biomolecules [Luo and Shoichet *Nature Materials*, 2004, 3, 249]; and sequential patterning and release of biomolecules [DeForest and Anseth *Angewandte Chemie International Edition*, 2012, 51, 1816].

3. Throughout the Results, the authors should include some motivating sentences to help guide the reader through the main points of each section.

4. Have the authors considered the attenuation of the light (for single photon exposure) through the z-dimension? Has this been characterized? Simple models exist to describe this for nitrobenzyl containing materials [Tibbitt et al. *Journal of Polymer Science Part A: Polymer Chemistry* 2013, 51, 1899; Norris et al. *Macromolecular Theory and Simulations* 2017, 26, 1700007].

5. The authors should include some rheology of the materials with photoresponsive cross-links to monitor the reaction kinetics in the system.

6. Further, the SEM images in Figure 5, should be replaced with profilometry (physical or optical) to characterize the state of the gel in the hydrated form.

7. The data included in Figure 6 is quite nice and could be highlighted even further. This is a clever use of the technology and shows how concepts from DNA gels [Cangialosi, Yoon et al. *Science* 2017, 357, 1126] can be extended with photoresponsive concepts.

8. There should be additional information about MUC-1 aptamers for a general audience.

9. The data in Figure 8 should be improved. The whole images should be shown instead of crops from several separate images. Further, a quantification of the cell numbers in the patterned regions as opposed to on the rest of the surface should be included.

Reviewer #3:

Remarks to the Author:

In this paper, Huang et al present the photopatterning of DNA based hydrogel matrices for cell adhesion and proliferation. In general there is much hope for DNA hydrogels to revolutionize ECM type materials for cell growth due to their extraordinary programmability in space, time, regulatory etc.

The study provides a good description of the structure of the hydrogel matrices and the photopatterning processes. 2D and 3D patterning of the photoresponsive hydrogel films are shown. The paper goes into great detail in explaining the fabrication techniques used to achieve photopattern in the hydrogel films. However, there is no clear synergy at present. The patterning techniques on DNA hydrogels are largely known (Few examples: 1. A. Cangialosi, C. Yoon, J. Liu, Q. Huang, J. Guo, T. D. Nguyen, D. H. Gracias and R. Schulman, *Science*, 2017, 357, 1126-1130. 2. P. J. Dorsey, M. Rubanov, W. Wang and R. Schulman, *ACS Macro Letters*, 2019, 8, 1133-1140). The patterning of cell adhesive patches to grow cells spatially selectively has been done many times and is rather routine – even if “fancy” here with DNA and HCR polymerization. At

present the paper does not offer real synergy from merging the two fields and that precludes a top level publication on the level of Nature communications.

Some minor point:

- Figure 1A caption should be more detailed. A lot of information is left in the text and not in the caption. This forces the reader go back and forth between the figures. Other figures are well made and sufficiently discussed.
- The paper introduces itself as one that designs a material to mimic the Extra Cellular Matrix (ECM). Mechanical information is however paramount for cell behavior. Apart from cell binding and growth, experiments regarding stress relaxation and hydrogel response to forces exerted by the cells are not shown. Essentially, we learn very little about the mechanical properties, and although fluorescence images are nice to look at, the tuning of the mechanical properties would be much more important. These properties are essential in material design to mimic ECM and it should be shown that how the hydrogels behavior.
- The authors attempt to show the effect of strain changes within the hydrogel matrix with confocal fluorescence microscope images. The hydrogels films did swell in respond to strain but this does not sufficiently prove the ECM mimicking abilities.

Re: Nature Communications Ms. NCOMMS-20-40528

Title: *"Spatiotemporal Patterning of Photoresponsive DNA-Based Hydrogels – Functional Matrices for Dictated Three-Dimensional Growth of Cells"*

Attached please find the corrected paper that addresses point-by-point the comments of the reviewers. The following changes were introduced into the paper (changes/additions marked in yellow):

Reviewer #1:

We appreciate the general comment of the reviewer "*The study is well-executed and the manuscript is well written. It will be of interest to a broad scientific community.*" The following changes address the specific comments:

1. *"The photocleavable linkage adopted here is similar to the one reported previously by the authors (Nano Lett. 2019, 19, 1, 618–625) and should be cited."*

Response: The reference, #68, was added, as requested.

2. *"The manuscript is entitled "spatiotemporal patterning hydrogels ... for dictated growth of cells". The term "dictation" could also indicate that the spatiotemporal variation of structure affects the cells behavior (e.g., dictation, migration, proliferation, etc.), which is not shown in the manuscript. Perhaps "dictated 3D growth" may be more suitable."*

Response: The suggestion of the reviewer to slightly alter the title of the paper was followed. The new title is "Spatiotemporal Patterning of Photoresponsive DNA-Based Hydrogels – Functional Matrices for Dictated Three-Dimensional Growth of Cells".

3. *"Will it be possible to comment on the maximum depth profile that can be achieved with the photopatterning?"*

Response: The depth profiling of the patterned circular domains was addressed twice in the corrected paper:

- a) On p.10 (end) and p.11, we explain that upon patterning, the UV light penetrates across the entire z-dimension of the thin film (80 μm), and thus, the profile of the patterns corresponds to the film thickness.
- b) On p. 17 we discuss the optical profiling of the patterned domains by reconstruction of the x, y, z fluorescent images and the DIC pattern, demonstrating the three-dimensional interior of the holes, generated across the film. A video movie that follows the reconstructed structure is provided.

4. *"Fig. 5: Can the pattern development into orthogonally arranged ellipses also be shown via SEM? Could denser areas between pores be created?"*

Response: Indeed, the orthogonally arranged ellipsoid structures could be followed by SEM. The SEM image is presented in Figure S5, and the result is, also, mentioned in the text, p. 21.

5. *“The 3D growth of cells in the photopatterned gel is convincing but the authors should also prove biocompatibility (e.g., live/dead cell staining). Live cell staining will also be valuable to provide the 3D distribution of living cells inside the holes.”*

Response: The biocompatibility issue of the cells in the patterned domains is addressed in the text, p. 25, and the accompanying image of the stained cells, probing live/dead cells, are provided in the new figure, Figure S11. The results demonstrate that, basically, all grown cells are alive and no cell motility could be detected. The three-dimensional distribution of the cells in the circular containments is presented in Figure S11.

6. a): *“Please provide the statistical analysis and error bars. Also include all experimental details (e.g., cell concentrations used for seeding and adsorption to the patterned areas).”*

Response: A statistical error bar analysis of the images is impossible. We explained in the experimental section that each of the photolithographic patterning protocols was applied in N=3-4 experiments and that no noticeable differences could be identified in the resulting patterns. The concentration of the seeded cells in the patterned domains (1×10^7 cells in 1 mL) was included to the experimental section, p. 32.

b): *“Why was 20% (w/w) acrylamide gels when performing the cell assays but 10% (w/w) for the rest of the studies?”*

Response: The reason for altering the concentration of the gel to 20% in the cell experiment was explained in the experimental section, p. 32.

7. Minor issues:

a): *“Is there any effect of the periodicity of the patterning on cell growth? It will be valuable if the authors can comment on this aspect.”*

Response: The spatial periodic separation of the patterns, their sizes and the surrounding stiffness of the hydrogel is, indeed, anticipated to influence the shape of the 3D cell aggregates. This issue is discussed on p. 29, and appropriate references, #78 and #79 are cited. Further future effects to alter the shapes of the grown cells by switchable stiffness of the hydrogel matrix by G-quadruplexes or T-A·T triplexes are discussed in the conclusion paragraph.

b): *“Fig. 4: It will be valuable if the authors can comment whether the signals be switched off again e.g. by competitive strands (also shown in earlier works). In addition, what is the minimum distance between the ellipses?”*

Response: The orthogonal ellipsoid pattern cannot be switched by a strand displacement process, due to the slow and inefficient displacement process. This is, however, a very interesting comment. In the conclusion paragraph we suggest to apply HCR transformation that guide the formation of reversible G-quadruplexes or T-A·T bridging units. These reversible bridges could switch the orthogonal ellipsoid structures.

c): *“p.13 l.303, red comma”*

Response: The typo mistake was corrected.

d): “Data on ‘programmed intercommunicating’ is missing (page 24, line 549-551).”

Response: The communication of cells is a future challenge, far beyond the scope of the present study. The word “**future**” was specifically mentioned for this possibility.

Reviewer #2:

We appreciate the comments of the review that “*The manuscript highlights a clever use of o-nitrobenzylphosphate esters to generate photoresponsive DNA-based hydrogels. The chemistry is nicely executed and the materials are described well. The results and possibilities are interesting.*” The comments of the reviewer were addressed as follows:

1. “*In the Abstract / Introduction, the specific contribution of this work is not made clear. There are general discussions about photoresponsive materials (many of which already exist) and the specific focus of this work is only evident later in the text. The authors should help guide the reader to the unique aspects of this work and better frame it within the context of the broader field.*”

Response: We followed the reviewer’s comment that the abstract and introduction sections insufficiently present the unique features of our photolithographic patterning approach and the fact that these parts did not emphasize the important accomplishment of the study. Accordingly, the abstract was rewritten to emphasize the specific contributions of this work. Also, we emphasized in the introduction the significance of the study and the broader impact of the results.

2. “*In this context, there are several research topics from the field of photoresponsive hydrogels (for biofunctionalization or cell culture) that should be acknowledged. There are much broader use of nitrobenzyl compounds in the design of photoresponsive gels.*”

Response: The references mentioned by the reviewer are, indeed, important and related to our study. The references, #43 to #47, were added into the introduction, and their relevance to the study was detailed.

3. “*Throughout the Results, the authors should include some motivating sentences to help guide the reader through the main points of each section.*”

Response: The comment of the reviewer was followed by describing the specific accomplishments and significance of each of the patterning platforms. This included a final statement summarizing the results for each of the systems.

4. “*Have the authors considered the attenuation of the light (for single photon exposure) through the z-dimension? Has this been characterized? Simple models exist to describe this for nitrobenzyl containing materials [Tibbitt et al. Journal of Polymer Science Part A: Polymer Chemistry 2013, 51, 1899; Norris et al. Macromolecular Theory and Simulations 2017, 26, 1700007].*”

Response: Indeed, the attenuation of the light plays an important role on the patterning quality due to the limited diffusibility of the degraded photoproducts. Nonetheless this issue is important for thick hydrogels revealing limited product diffusibility. In our case, we use very thin films (80 μm) of hydrated hydrogels, where the diffusion of waste products to the bulk solution is not limited. We discussed the issue of light-attenuation effects on the patterning of orthonitrobenzyl ester protected hydrogel, p.6, added the appropriate reference, #76 and #77, to the paper, and explained that light-attenuation effects on our thin film is negligible.

5. *“The authors should include some rheology of the materials with photoresponsive cross-links to monitor the reaction kinetics in the system.”*

Response: Rheology results characterizing the stiffness properties of the photodeprotected hydrated hydrogels were added, Figure S3, and the results were discussed in the text.

6. *“Further, the SEM images in Figure 5, should be replaced with profilometry (physical or optical) to characterize the state of the gel in the hydrated form.”*

Response: The SEM image that was in the original manuscript was transferred to the supporting information. We present in Figure 5(C) new optical profiling of the holes by the reconstruction of the x, y, z confocal fluorescence images. The reconstruction reveals the 3D cavity domains of the pattern. In addition, we added to the supporting information a video that shows the reconstructed cavity image that demonstrate clearly its profile.

7. *“The data included in Figure 6 is quite nice and could be highlighted even further. This is a clever use of the technology and shows how concepts from DNA gels [Cangialosi, Yoon et al. Science 2017, 357, 1126] can be extended with photoresponsive concepts.”*

Response: Indeed, we appreciate the identification of the results shown in Figure 6 as a key observation of the study. The application of DNA-based hydrogels for controlling shapes and structures of patterns is, indeed, important. The reference mentioned by the reviewer was included in the original text. We added, however, an additional reference #80 that reflects the significance of DNA-based hydrogels on patterned domains. In addition, we introduced into the conclusion paragraph a short discussion addressing the future application of switchable DNA hydrogels of patterned shapes. This discussion complements the reviewer’s comment regarding the significance of coupling DNA nanotechnology concepts to photoresponsive materials.

8. *“There should be additional information about MUC-1 aptamers for a general audience.”*

Response: Additional information on the MUC-1 aptamer was introduced into the text, p. 24.

9. *“The data in Figure 8 should be improved. The whole images should be shown instead of crops from several separate images. Further, a quantification of the cell*

numbers in the patterned regions as opposed to on the rest of the surface should be included.”

Response: Figure 8, includes new images that emphasize that all calcein-AM stained alive cells are confined to the patterned domains. The results are presented in the new Figure 8, as requested by the reviewer.

Reviewer #3:

The reviewer is, certainly, correct that different photopatterning methods of hydrogels were reported and that cells were positioned, and selectively grown, on selective patches of hydrogels. We feel, however, that the reviewer missed some important results introduced by the study. In the corrected paper, we tried to emphasize these points:

- a) The present study introduces a common approach to pattern hydrogel matrices by versatile combinations of o-nitrobenzylphosphate ester photoprotective units. This allows the preparation of a universal set of patterned materials revealing guided functionalities and programmed stiffness properties. This issue was further emphasized in the introduction.
- b) The demonstration that the stress-induced hybridization chain reaction (HCR) of the bulk hydrogel controls the shapes of the spatially separated domains, leading to the orthogonal ellipsoid structures, is a significant result. The reviewer is correct that previous studies demonstrated that the stiffness of hydrogels (due to controlled crosslinking) affects the shapes of patterned domains (and the references mentioned by the reviewer are present in the paper), yet the HCR approach demonstrates how principle of DNA nanotechnology can be adapted to control the shapes of the patterns. This HCR principle paves new versatile methods to control the stiffness and switch the stiffness, thereby allowing to design new stress interactions between the hydrogel and low-stiffness domain and their loads. This point is further emphasized in the text, p. 21, and in the conclusion paragraph p. 30.
- c) The review remark that we did not present force interactions between the HCR-induced stress and the cell-loaded domains is certainly correct. In the corrected paper, we describe rheology experiments probing the stiffness of the hydrogel before and after the HCR. Particularly, we follow by rheometry the effects of the grown cells on the stiffness of the hydrated hydrogel, thereby demonstrating force interactions between the cells and the surrounding hydrogel.

The specific comments of the reviewer were addressed as follows:

1. *“Figure 1A caption should be more detailed. A lot of information is left in the text and not in the caption. This forces the reader go back and forth between the figures. Other figures are well made and sufficiently discussed.”*

Response: The content of the figure caption was further detailed, as requested.

2. *“The paper introduces itself as one that designs a material to mimic the Extra Cellular Matrix (ECM). Mechanical information is however paramount for cell behavior. Apart from cell binding and growth, experiments regarding stress*

relaxation and hydrogel response to forces exerted by the cells are not shown. Essentially, we learn very little about the mechanical properties, and although fluorescence images are nice to look at, the tuning of the mechanical properties would be much more important. These properties are essential in material design to mimic ECM and it should be shown that how the hydrogels behavior.”

Response: As stated, we describe in the corrected paper rheology experiment that follows the interaction between the cells and the stiffness properties of the surrounding hydrogel (p. 24).

3. *“The authors attempt to show the effect of strain changes within the hydrogel matrix with confocal fluorescence microscope images. The hydrogels films did swell in respond to strain but this does not sufficiently prove the ECM mimicking abilities.”*

Response: In addition to confocal microscopy images that probe ECM mimicking abilities, we add now the SEM image of the hydrated hydrogel, Figure S5, the reconstruction of the patterned patches, Figure 5(C). We, also, provide the video of the reconstructed patterned hydrogel, movie S3.

We believe that with these corrections and explanations we addressed the comments of the reviewers.

We thank and appreciate the valuable comments of the reviewers, and the suggestions introduced by the editor.

I look forward to the publication of the paper in Nature Communications.

Sincerely Yours

Prof. Itamar Willner

Reviewers' Comments:

Reviewer #1:

Remarks to the Author:

The authors have addressed the concerns adequately and the manuscript is suitable for publication

Reviewer #2:

Remarks to the Author:

The authors have adequately addressed the main points in the revised version of the manuscript and this interesting application of photoresponsive materials is now suitable for publication.

Reviewer #3:

Remarks to the Author:

The revised version has properly addressed some reviewer questions. However, as mentioned in my previous statement, I do not think that this manuscript is on the level of Nature Communication. Many aspects of the patterning are known and the cell work is largely disconnected from the hydrogels and on a very preliminary level. Why are the hydrogels needed to seed the cells on a hydrogel surface and selectively grow them on the bioactive spot? It could just be done on a MUC surface with almost exactly the same result. all the DNA chemistry underneath has almost no effect on the cells except of making them attach to a spot.

The most noteworthy result of the whole paper is the formation of ordered patterns by growth of the spots by hairpin polymerization. This is remarkable. Very little detail is however presented. This growth is also performed under the cell layers, but again, very little data is presented. Can the cells proliferate more strongly? Can the cell number be increased by increasing the spot size via the growth? Is there any mechanotransduction? Many questions could be asked, but we mostly see end point data. At the end this is an editorial decision. It is good and solid science, but for me the cell work is just an add on without any clear benefit.

One thing which must be corrected in any case is the newly introduced claim on 3D Growth of cells. Patterning cell layers on lateral patterns is not 3D. There is not a single piece of data to support in-growth and proliferation of the cells within the posts.